# Elevated Intrarenal Resistive Index Predicted Faster Renal Function Decline and Long-Term Mortality in Non-Proteinuric Chronic Kidney Disease

**DOI:** 10.3390/jcm11112995

**Published:** 2022-05-25

**Authors:** Giulio Romano, Roberto Mioni, Nicola Danieli, Martina Bertoni, Elisa Croatto, Lucia Merla, Lucia Alcaro, Antonio Pedduzza, Xenia Metcalf, Alessandra Rigamonti, Cristiana Catena, Leonardo A. Sechi, GianLuca Colussi

**Affiliations:** 1Nephrology, Department of Medicine, University of Udine, 33100 Udine, Italy; giulio.romano@uniud.it (G.R.); danieli-nicola@hotmail.it (N.D.); bertoni.martina@spes.uniud.it (M.B.); 2Division of Nephrology, Academic Hospital of Udine “Santa Maria della Misericordia”, 33100 Udine, Italy; roberto.mioni@asufc.sanita.fvg.it; 3Internal Medicine, Department of Medicine, University of Udine, 33100 Udine, Italy; elisa.croatto@gmail.com (E.C.); luciamerla@gmail.com (L.M.); alcaro.lucia@spes.uniud.it (L.A.); pedduzza.antonio@gmail.com (A.P.); dottssametcalf@gmail.com (X.M.); cristiana.catena@uniud.it (C.C.); leonardo.sechi@uniud.it (L.A.S.); 4Department of Radiology, Academic Hospital of Udine “Santa Maria della Misericorida”, 33100 Udine, Italy; alessandra.rigamonti@asufc.sanita.fvg.it

**Keywords:** atherosclerosis, renal artery stenosis, renal failure, vascular resistance

## Abstract

Background. Intrarenal resistive index (RI) ≥ 0.80 predicts renal outcomes in proteinuric chronic kidney disease (CKD). However, this evidence in non-proteinuric patients with CKD of unknown etiology is lacking. In this study, we assessed the effect of intrarenal RI on renal function and all-cause mortality in non-proteinuric patients with CKD of unknown etiology despite an extensive diagnostic work-up. Methods. Non-proteinuric CKD patients were evaluated in a retrospective longitudinal study. Progression of renal disease was investigated by checking serum creatinine levels at 1, 3, and 5 years and defined by a creatinine level increase of at least 0.5 mg/dL. The discrimination performance of intrarenal RI in predicting the 5-year progression of renal disease was assessed by calculating the area under the receiver operating characteristic curve (AUROC). Results. One-hundred-thirty-one patients (76 ± 9 years, 56% males) were included. The median follow-up was 7.5 years (interquartile range 4.3–10.5) with a cumulative mortality of 53%, and 5-year renal disease progression occurred in 25%. Patients with intrarenal RI ≥ 0.80 had a faster increase of serum creatinine levels compared to those with RI < 0.80 (+0.06 mg/dL each year, 95% CI 0.02–0.10, *p* < 0.010). Each 0.1-unit increment of intrarenal RI was an independent determinant of 5-year renal disease progression (odds ratio 4.13, 95% CI 1.45–12.9, *p* = 0.010) and predictor of mortality (hazards ratio 1.80, 95% CI 1.05–3.09, *p* = 0.034). AUROCs of intrarenal RI for predicting 5-year renal disease progression and mortality were 0.66 (95% CI 0.57–0.76) and 0.67 (95% CI 0.58–0.74), respectively. Conclusions. In non-proteinuric patients with CKD of unknown etiology, increased intrarenal RI predicted both a faster decline in renal function and higher long-term mortality, but as a single marker, it showed poor discrimination performance.

## 1. Introduction

In order to prevent the progression of CKD to end-stage renal disease (ESRD), it is essential to identify the etiology of the underlying renal disease. Non-proteinuric CKD is common in patients who develop ESRD [1] and could represent most of the 20% of CKD cases in which the origin cannot be found despite an extensive diagnostic evaluation (CKD of unknown etiology) [2].

Renovascular disease is a complex disorder and comprises chronic renal ischemia induced by anatomical alterations in the renal arteries that may reduce renal perfusion. The most frequent cause of these vascular abnormalities is related to atherosclerosis [3]. Renovascular disease usually presents in two different clinical forms: Renovascular Hypertension, characterized by renal artery stenosis, systemic hypertension, and normal renal function; Ischemic Nephropathy, characterized by renal artery stenosis, increased intrarenal vascular resistance, and a decrease in renal function with or without hypertension [3]. Renal failure because of ischemic nephropathy results from functional, hormonal, and anatomical alterations induced by chronic renal hypoperfusion and ischemia [4]. Currently, the hypothesis that an increased intraparenchymal vascular resistance without renal artery stenosis may cause glomerular hypoperfusion and a decline in glomerular filtration rate towards ESRD, not associated with glomerular damage, has not been well defined [5].

Atherosclerotic risk factors are common in patients with non-proteinuric CKD [6] and atherosclerosis has been associated with renal vascular dysfunction [7] and progression of renal disease [8,9]. Although treatment of atherosclerotic risk factors can reduce the probability of CKD progression, in some cases, the renal function can decline anyway. In particular, in these cases, the persistence of residual atherosclerotic risk and structured vascular damage can induce a vicious circle in which the inflammation of the vascular wall promotes a systemic endothelial dysfunction with further atherosclerotic damage and renal function decline [10,11]. It has been shown that CKD is associated with abnormalities in intrarenal [12], coronary [13], and carotid arteries [11], as well as with the reduction of the blood flow in the cerebral arteries [14]. This systemic vascular dysfunction associated with renal function decline handles the elevated risk of cardiovascular events and mortality in patients with CKD of any etiology [9].

The intrarenal resistive index (RI) is a marker of both intrarenal vascular dysfunction and systemic atherosclerosis [15,16] and has shown promising results as a predictor of renal function decline and all-cause mortality after renal angioplasty and in proteinuric patients [17,18]. Nevertheless, the predictive role of intrarenal RI in the less common setting of non-proteinuric patients with CKD of unknown etiology is lacking. The aim of this study was to evaluate the role of intrarenal RI in predicting renal function decline and all-cause mortality in non-proteinuric patients with CKD of unknown etiology despite an extensive diagnostic work-up.

## 2. Patients, Materials, and Methods

### 2.1. Study Design

Retrospectively, we evaluated an initial cohort of 1563 consecutive patients admitted to the Nephrology Outpatient Clinics of the University of Udine (Italy) because of CKD from June 2004 to November 2017. Patients were then followed up in the Nephrology or Internal Medicine clinic up to January 2022. We selected patients older than 18 years, of all sexes, with a creatinine clearance between 89 and 15 mL/min/1.73 m^2^, and before they started any treatment with angiotensin-converting-enzyme inhibitors (ACEi) or angiotensin receptor blockers (ARB). We excluded from the initial cohort patients who showed signs of active renal disease or with a medical history of kidney disease, active hematologic or solid neoplasia, polycystic kidney disease, or renal artery stenosis. We considered active renal disease the presence of albuminuria or proteinuria equal to or higher than 30 or 200 mg/day, respectively, hematuria of glomerular origin by polarized light microscopy, cellular casts, or signs of urinary tract infection at urinalysis. Ultrasonographic exclusion criteria were dilatation of the renal calyxes or hydronephrosis and a difference of over one centimeter between left and right maximum longitudinal kidney length. Renal artery stenosis was suspected by a peak-systolic velocity (PSV) ≥ 180 cm/s or a difference between right and left mean RI higher than 5% by color Doppler analysis [19]. In each patient, we collected information about the history of past or active smoking, hypertension, dyslipidemia, diabetes mellitus, coronary artery disease, atrial fibrillation, heart failure, transitory ischemic attack, stroke, and peripheral artery disease. The body mass index (BMI) was calculated as body weight (kilograms) divided by the square of height (meters). Blood pressure was measured during the first visit after 15-min rest in a sitting position. Systolic and diastolic blood pressure (SBP and DBP, respectively) were assessed in the dominant arm with an automated oscillometric sphygmomanometer (M2 HEM-7121-E, OMRON, Kyoto, Japan). All patients performed general laboratory blood and urine tests by standard methods. Collected biochemical variables were plasma total cholesterol, low-density lipoprotein (LDL) cholesterol, high-density lipoprotein (HDL) cholesterol, triglycerides, and glucose levels. The non-HDL cholesterol was calculated as an atherogenic marker by subtracting HDL cholesterol from the total cholesterol levels as an atherogenic marker [20]. Daily excretion of albumin and total proteins was measured in a 24-h urine collection. Serum creatinine was measured to monitor in each patient the progression of renal disease at 1, 3, and 5 years after the first medical visit. Renal disease progression over 5 years was defined by a creatinine increment of at least 0.5 mg/dL. Mortality was assessed by checking the accessible hospital electronic record of each patient, up to 31 January 2022. Laboratory tests and imaging were performed in the Division of Laboratory Medicine and the Institute of Radiology of the Academic Hospital of Udine. Patients’ data were collected from hospital clinical records and anonymized before including them in the database.

This study was performed according to principles established in the Declaration of Helsinki. All patients read the consent document and gave their informed consent for using clinical data for research at the first visit. Data were anonymized before they were stored in the database. The Institutional Review Board of the University of Udine approved the study protocol (protocol number 02/IRB-ROMANO_2019) and stated that any additional specific informed consent for the retrospective analysis of data was unnecessary.

### 2.2. Renal Function and Ultrasonographic Examination

Glomerular filtration rate (GFR) was calculated by using the creatinine clearance measured on 24-h urine collection adjusted for 1.73 m^2^ body surface area. A renal ultrasound examination was performed with a duplex Doppler apparatus after a 12-h fasting period, as previously described [21]. Briefly, a 3.5-MHz convex phased-array probe and a color-Doppler mapping were used to identify the renal arteries. The Doppler angle was kept lower than 60° and close as possible to zero to measure PSV and end-diastolic velocity (EDV) at the level of renal arteries and in the interlobar arteries next to medullary pyramids (intrarenal medium-sized arteries) [22]. The intrarenal RI was calculated according to the formula (PSV-EDV)/PSV, where PSV and EDV were measured in cm/s at the superior, medium, and inferior regions of both kidneys and expressed as the mean value of all measures. According to Radermacher et al., an RI ≥ 0.80 was predictive of renal disease progression and mortality [18] and 0.80 was used as a reference cutoff also in our study. Left and right longitudinal kidney dimensions and the cortical thickness were measured in multiple longitudinal images obtained in sagittal and coronal planes and were reported as mean values of the two kidneys. Intra- and inter-observer coefficients of variability for intrarenal RI measurement by color-Doppler ultrasound were previously reported [21].

### 2.3. Statistical Analysis

Data were reported as mean ± standard deviation for the normal variables or as the median and interquartile range (IQR) for the skewed ones. Variables not normally distributed were log-transformed before analysis with parametric methods. Countable variables were reported as proportions in contingency tables. The difference between means was assessed by Student’s *t*-test and between proportions by Fisher’s exact test. Correction for multiple comparisons was performed with the Tukey method. Correlation analysis was performed by calculating Pearson or Spearman correlation coefficient (*r*) for normal or skewed distributed variables, respectively. Cross-sectional univariate and multivariate analyzes were performed by linear regression with the ordinary least-square method. Serum creatinine variation as a marker of renal disease progression was assessed across 5 years according to baseline intrarenal RI ≥ 0.80 or <0.80 by mixed-effect linear regression. In the multivariate mixed effect model, continuous variables were standardized (scaled) by subtracting each value from the mean and dividing the result by the standard deviation. Mixed effect estimates were reported as standard coefficients with a 95% confidence interval (CI). Progression of renal disease was assessed also as a dummy variable by determining an increment of creatinine levels of at least 0.5 mg/dL during the 5-year follow-up. Variables associated with renal disease progression, as a dummy variable, were analyzed by logistic regression and expressed as odds ratio (OR) with 95% CI. Survival analysis of patients with an RI ≥ 0.80 or <0.80 is presented with the Kaplan-Meier curves and analyzed with the log-rank statistic. Predictors of mortality were assessed by Cox proportional hazards regression and expressed as hazards ratio (HR) and 95% CI. The best multivariate prediction models were determined by stepwise forward-backward variable selection according to the Akaike Information Criterion. The discrimination performance of intrarenal RI as a predictor of 5-year renal disease progression and long-term mortality was determined by calculating the area under the receiver operating characteristic curve (AUROC). We estimated also sensitivity, specificity, and positive and negative predictive values of the reference intrarenal RI cutoff value of 0.80 for both outcomes. The null hypothesis of all statistical tests was rejected when the probability (*p*) of accepting it was lower than 5% (*p* < 0.050). Statistical analysis was performed with the free software R (version 4.1.1, R Core Team, Vienna, Austria) [23] with survival (version 3.3-1, Therneau TM, Mayo Clinic, Rochester, MN, USA) [24], lmerTest (version 3.1-3, Kuznetsova et al., Technical University of Denmark, Lyngby, Denmark) [25], and cutpointr (version 1.1.2, Thiele and Hirschfeld, University of Applied Sciences, Bielefeld, Germany) [26] packages.

## 3. Results

After applying exclusion criteria, 131 patients remained eligible for the study (Figure 1). The mean age of patients was 76 ± 9 years, 56% were males, and 27% were past or active smokers. The most frequent comorbidities were 89% hypertension, 28% diabetes mellitus, 16% heart failure, 15% transitory ischemic attack or stroke, 14% coronary artery disease, 14% atrial fibrillation, and 7% peripheral artery disease. Statins were used by 55% of patients. Evaluation of the baseline renal function showed a serum creatinine level of 1.6 ± 0.4 mg/dL, GFR 47 ± 16 mL/min/1.73m^2^, 24-h urinary albumin excretion 23 mg/day (IQR 7–29), and 24-h urinary protein excretion 120 mg/day (IQR 74–154). The longitudinal kidney diameter was 10.1 ± 1.2 mm, the cortical thickness was 1.4 ± 0.2 mm, and the intrarenal RI was 0.77 ± 0.05. Intrarenal RI ≥ 0.80 was documented in 39 patients. The median follow-up time of the study was 7.5 years (IQR 4.3–10.5) and 70 patients (53%) deceased within this period.

### 3.1. Baseline Variable Analysis

Variables positively correlated to baseline GFR were longitudinal kidney diameter and kidney cortical thickness, whereas those negatively correlated were age, serum creatinine levels, and intrarenal RI (Table 1). The best model of variables associated with GFR comprised age, SBP, total cholesterol, longitudinal kidney diameter, kidney cortical thickness, and intrarenal RI (Table 1). Variables positively associated with intrarenal RI were age, use of statins, and serum creatinine level, whereas those negatively associated were total cholesterol, non-HDL cholesterol, and GFR (Table 1). The best model of variables associated with intrarenal RI comprised non-HDL cholesterol, GFR, and kidney cortical thickness (Table 1). Both total cholesterol and non-HDL cholesterol correlated negatively with statin use, *r* = −0.730 (*p* < 0.001) and *r* = −0.732 (*p* < 0.001), respectively. Patients with an intrarenal RI ≥ 0.80 had a higher proportion of heart failure, use of statins, and levels of serum creatinine, whereas they had lower levels of total cholesterol, non-HDL cholesterol, and GFR compared to patients with RI < 0.80 (Table 2).

### 3.2. Renal Function Longitudinal Analysis

Serum creatinine levels of all patients collected during the 5 years of follow-up correlated positively with intrarenal RI (Figure 2) and serum creatinine levels increased over time in all patients (0.04 mg/dL each year, 95% CI 0.02–0.06, *p* < 0.001). When variables intrarenal RI and time, and their interaction term (RI × time) were included in the regression model, serum creatinine increased by 0.02 mg/dL each year. Patients with RI ≥ 0.80 had a creatinine concentration 0.28 mg/dL higher than those with RI < 0.80, and creatinine levels increased each year by 0.06 mg/dL more in patients with RI ≥ 0.80 compared to those with RI < 0.80 (model 1, Table 3). The interaction term remained statistically significant after adjusting the model for age and sex (model 2, Table 3), and baseline GFR (model 3, Table 3). The faster rate of progression of serum creatinine levels in patients with RI ≥ 0.80 compared to the rate of progression of creatinine in patients with RI < 0.80 is presented in Figure 3. The cumulative incidence of 5-year renal disease progression was 25%. Variables associated with 5-year renal disease progression in the univariate logistic analysis were coronary artery disease, heart failure, atrial fibrillation, 24 h-urinary albumin, 24 h-urinary protein, and intrarenal RI. Variables independently associated with 5-year renal disease progression in the multivariate best model were coronary artery disease, atrial fibrillation, total cholesterol levels, and intrarenal RI (Table 4). The AUROC of intrarenal RI as a predictor of 5-year renal disease progression was 0.66 (95% CI 0.57–0.76) and the reference RI cutoff of 0.80 showed 48% sensitivity (95% CI 31–66), 81% specificity (95% CI 71–88), 46% positive predictive value (95% CI 29–63), and 82% negative predictive value (95% CI 73–89).

### 3.3. Survival Analysis

Deceased patients had a median follow-up of 4.7 years (IQR 3.1–8.3) and those censored 8.7 years (IQR 7.3–13). Of the 61 censored patients, seven were lost during the follow-up, and the others were alive at the last control. Deceased patients were older, and had higher intrarenal RI, lower BMI, and lower GFR compared to those censored (Table 2). The proportion of intrarenal RI ≥ 0.80 was 36% in deceased patients and 16% in those censored (*p* = 0.017). In Figure 4, we present the Kaplan-Meier curves that show a higher survival probability in patients with baseline intrarenal RI < 0.80 than those with RI ≥ 0.80. The median survival time in patients with RI ≥ 0.80 was 7.7 years, and that of patients with RI < 0.80 was 11.1 years. Univariate predictors of mortality were coronary artery disease and heart failure, higher serum creatinine levels and intrarenal RI, and lower BMI and GFR. The best prediction model of mortality comprised the variables age, diabetes, coronary artery disease, heart failure, BMI, glycemia, and intrarenal RI (Table 5). The AUROC of intrarenal RI as a predictor of mortality was 0.67 (95% CI 0.58–0.74) and the reference RI cutoff of 0.80 showed 36% sensitivity (95% CI 25–48), 84% specificity (95% CI 72–92), 71% positive predictive value (95% CI 54–85), and 53% negative predictive value (95% CI 43–63).

## 4. Discussion

The predictive role of intrarenal RI for CKD progression and mortality was previously shown by Radermacher et al., in patients affected by the atherosclerotic renovascular disease [17] and in patients without renal artery stenosis but with proteinuric CKD [18]. In the first case, intrarenal RI < 0.80 predicted the response in terms of improvement of renal function, blood pressure control, and reduced need for renal replacement therapy after renal artery revascularization [17]. In the second, intrarenal RI ≥ 0.80 predicted the decline of renal function, the need for renal replacement therapy, and all-cause mortality [18]. Our study confirms the importance of intrarenal RI as a determinant of renal disease progression and mortality and extends Radermacher et al. observations to the new clinical setting of non-proteinuric patients with CKD of unknown etiology despite extensive diagnostic work-up.

The lack of a relationship between GFR and microalbuminuria in our study confirmed that CKD was not associated with overt glomerular damage. Therefore, the inverse relationship between GFR and intrarenal RI observed suggests glomerular hypoperfusion was the principal mechanism. According to the literature, whether glomerular hypoperfusion and increased RI are related to systemic hemodynamic factors or intrarenal structural abnormalities remains a matter of debate [27]. Some studies showed systemic hemodynamic factors influence intrarenal RI in hypertensive patients with and without CKD. Akaishi et al. observed that intrarenal RI is associated with diastolic to systolic blood pressure ratio [28]. Geraci et al. demonstrated that intrarenal RI is associated with pulse pressure and carotid intima-media thickness, and similar findings were documented by other authors [7]. Other studies highlighted the role of intrarenal abnormalities. The prospective study of Stefan et al., which included traditional cardiovascular risk factors as confounders, showed that intrarenal arteriosclerosis is the only independent predictor of increased RI in CKD patients [27]. Other authors observed also a strong and independent association between intrarenal RI, glomerulosclerosis, and tubulointerstitial damage [12,29]. Based on this evidence, we might hypothesize that the reason for glomerular hypoperfusion and increased intrarenal RI in our patients is a mixed contribution of both systemic hemodynamic factors and intrarenal abnormalities. Therefore, the better condition that could describe the non-proteinuric kidney dysfunction of unknown etiology is Atherosclerotic Nephropathy.

In several studies, increased intrarenal RI is directly correlated to cardiovascular risk, systemic atherosclerosis, and mortality. For example, in essential hypertensive patients with GFR higher than 30 mL/min/1.73 m^2^, Catena et al. showed that intrarenal RI is associated with a pro-thrombotic state [30], while Pontremoli et al. observed an independent correlation between intrarenal RI, carotid intima-media thickness, and left ventricular hypertrophy [15]. In addition, Watanabe et al. observed that intrarenal RI is associated with multi-site atherosclerosis assessed by the coronary artery calcium score and the carotid intima-media thickness in patients with preserved renal function and these associations are independent of hypertension [31]. Last, Toledo et al. showed that intrarenal RI ≥ 0.70 is associated with increased all-cause mortality, and observed a slight increase in cardiovascular mortality in patients with higher RI [32]. According to this evidence, also in our study, we observed that patients with RI ≥ 0.80 were slightly older and had a higher prevalence of male sex, smoking history, hypertension, diabetes, heart failure, atrial fibrillation, peripheral artery disease, and statins use. The elevated atherosclerotic burden on our patients might have justified the faster decline in renal function and the higher mortality rate that we have observed in those with the higher intrarenal RI.

Although in our study intrarenal RI was an independent predictor of 5-year renal disease progression and long-term mortality, the low AUROC and cutoff metrics for both outcomes denote a poor discrimination performance of this marker and suggest the importance of considering additional risk factors. In both our multivariate models for renal disease progression and mortality, we observed also the significant effect of classical cardiovascular risk factors and history of cardiovascular events. Including intrarenal RI and other relevant risk factors in a more complex prediction model should improve the model performance, but this goes beyond the present study.

In this study, we noticed an inverse relationship between intrarenal RI and cholesterol. This paradoxical association occurred probably because of the inverse association between cholesterol levels and the use of statins and can be explained by collider stratification bias [33]. Notably, in our study, patients with the higher intrarenal RI were more frequently treated with statins because of their elevated cardiovascular risk, but statins might influence the renal function [34] and act as statistical confounders. The role of statins in the progression of CKD based on intrarenal RI is not an object of this study, but the clinical relevance of this relationship would merit further studies.

The first limit of our study is the retrospective design because of the risk of introducing selection bias. However, from another point of view, the retrospective design has given us the opportunity to observe long-term outcomes that could not be observed in a prospective study because of time and resource limits. Second, although we included patients before they started any ACEi/ARB, these and other drugs influencing renal blood flow and cardiovascular risk might have been introduced in most of our patients during follow-up. Since this was not considered, we cannot exclude that our results were affected by different drugs administration. This is an important point since renal disease progression and cardiovascular risk is strongly influenced by the type of therapy taken and by medication adherence during follow-up. Third, this was a monocentric study, and our findings should be limited to cohorts of patients with similar characteristics. However, the advantage of our monocentric study is that patients were followed up in the same clinical setting so that a homogeneous diagnosis and treatment were granted to all patients during the study period.

In conclusion, this study showed that an elevated intrarenal RI predicts a faster decline in renal function and higher long-term mortality in non-proteinuric patients with CKD of unknown etiology despite extensive diagnostic work-up. Although our findings extend previous Radermacher et al. observations to another common clinical setting of CKD, intrarenal RI as a single marker of renal disease progression and mortality showed a poor discrimination performance. Further studies should consider the clinical utility of including intrarenal RI in more complex prediction models to assess renal outcomes and all-cause mortality and explore its potential role as a target for new preventive strategies in this particular CKD setting.

## Figures and Tables

**Figure 1 jcm-11-02995-f001:**
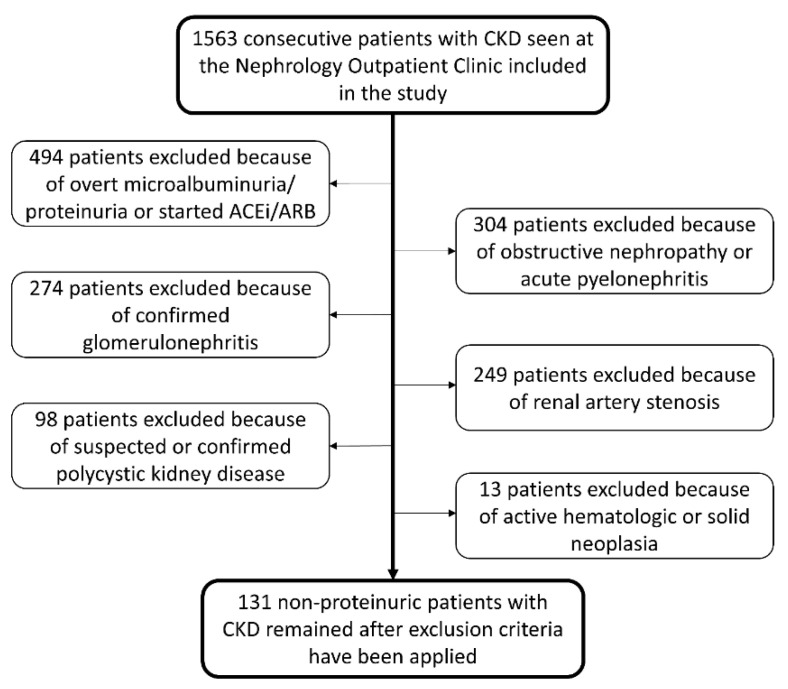
Flowchart of patients’ selection.

**Figure 2 jcm-11-02995-f002:**
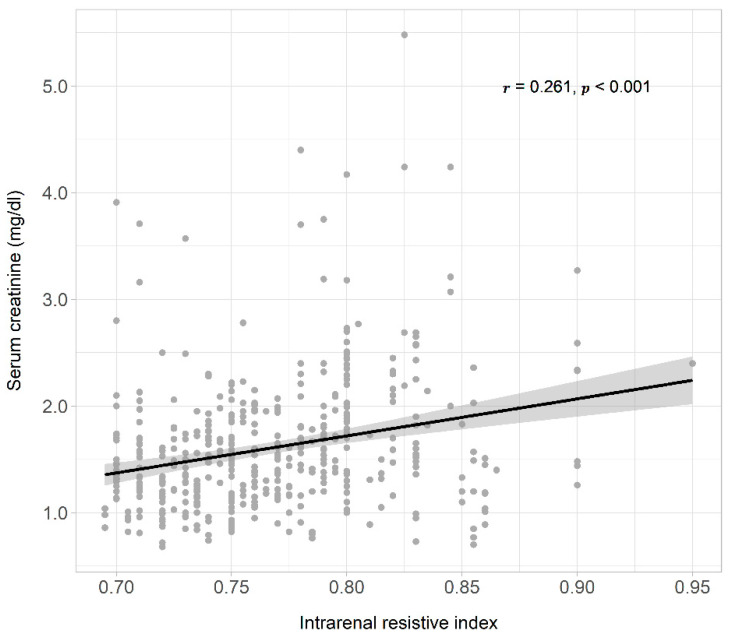
Scatter plot, regression line, and 95% confidence interval of the dependent variable, all serum creatinine levels measured during 5 years of observation, and the independent variable intrarenal resistive index. On the top right were reported the correlation coefficient (*r*), and the probability (*p*).

**Figure 3 jcm-11-02995-f003:**
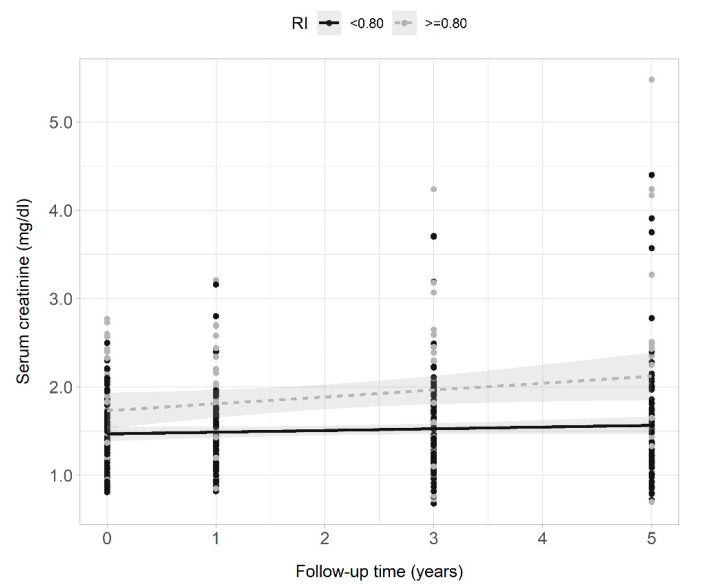
Scatter plots, regression lines, and 95% confidence intervals of the dependent variable serum creatinine clearance and the fixed factor time of patients with intrarenal RI ≥ 0.80 (gray points and dashed line) and <0.80 (black points and continuous line) across the 5 years of observation.

**Figure 4 jcm-11-02995-f004:**
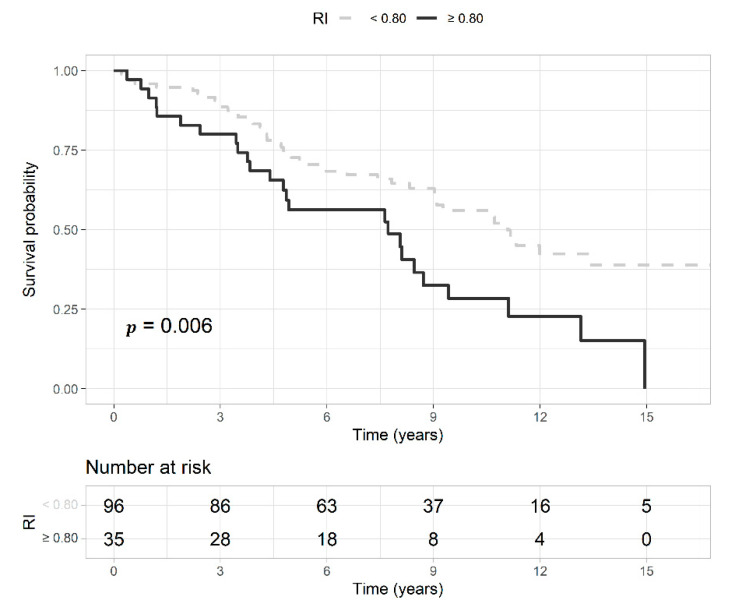
Kaplan-Meier curves represent the survival probability of patients with an intrarenal resistive index (RI) ≥ 0.80 (gray and dashed line) or <0.80 (black and continuous line) with the probability (*p*) of the log-rank test. Below the curves, is reported the risk table with the number of patients at risk at each time point.

**Table 1 jcm-11-02995-t001:** Baseline correlation analysis and best model of determinants of GFR and intrarenal resistive index.

	GFR		Intrarenal RI	
Variable	Correlation Coefficient	Best Model Standard Coefficient	Correlation Coefficient	Best Model Standard Coefficient
Age (years)	−0.313 ***	−0.192 *	0.175 *	-
Male sex (yes/no)	0.109	-	0.099	-
Past or active smoker (yes/no)	0.068	-	0.044	-
History of (yes/no): Hypertension Diabetes Coronary artery disease Heart failure Atrial fibrillation Tia/stroke Peripheral artery disease	−0.019 0.128 0.049 −0.120 −0.030 −0.064 −0.084	- - - - - - -	0.110 0.071 0.046 0.143 0.064 −0.018 0.133	- - - - - - -
Body mass index (kg/m^2^)	−0.054	-	0.011	-
Systolic blood pressure (mm Hg)	0.110	0.132	−0.081	-
Pulse pressure (mm Hg)	0.080	-	0.041	-
Plasma glucose levels (mg/dL)	−0.085	-	0.091	-
Total cholesterol (mg/dL)	−0.157	−0.187 *	−0.244 **	-
Non-HDL cholesterol (mg/dL)	−0.121	-	−0.270 **	−0.308 ***
Triglycerides (mg/dL)	0.041	-	−0.010	-
Statin user (yes/no)	0.045	-	0.254 **	-
Serum creatinine (mg/dL)	−0.545 ***	-	0.347 ***	-
Serum uric acid (mg/dL)	−0.120	-	0.098	-
GFR (mL/min/1.73 m^2^)	-	-	−0.299 ***	−0.383 ***
24-h urinary albumin (mg/day)	−0.002	-	0.005	-
24-h urinary protein (mg/day)	−0.014	-	0.014	-
Longitudinal kidney diameter (mm)	0.331 ***	0.175 *	0.051	-
Kidney cortical thickness (mm)	0.362 ***	0.233 **	0.008	0.127
Intrarenal RI	−0.299 ***	−0.311 ***	-	-

* *p* < 0.050; ** *p* < 0.010; *** *p* < 0.001; RI, resistive index; Tia, transitory ischemic attack; GFR, glomerular filtration rate by creatinine clearance.

**Table 2 jcm-11-02995-t002:** Variables description of patients with baseline intrarenal RI above or below the cut-off of 0.80 and between alive/censored or deceased at the end of follow-up.

	Intrarenal RI		Mortality Outcome	
Variable	<0.80 (*n* = 96)	≥0.80 (*n* = 35)	Censored (*n* = 61)	Deceased (*n* = 70)
Age (years)	75 ± 9	77 ± 9	72 ± 9	79 ± 7 ***
Male sex (*n* (%))	50 (52)	23 (66)	32 (53)	41 (59)
Past or active smoker (*n* (%))	18 (19)	9 (26)	12 (20)	15 (21)
Diagnosis of (*n* (%)): Hypertension Diabetes Coronary artery disease Heart failure Atrial fibrillation Tia/stroke Peripheral artery disease	84 (88) 25 (26) 13 (14) 10 (10) 10 (10) 15 (16) 4 (4.2)	33 (94) 12 (34) 5 (14) 11 (31) ** 8 (23) 5 (14) 5 (14)	56 (92) 18 (30) 5 (8.2) 6 (9.8) 6 (9.8) 8 (13) 2 (3.3)	61 (87) 19 (27) 13 (19) 15 (21) 12 (17) 12 (17) 7 (10)
Body mass index (kg/m^2^)	30.8 ± 5.8	30.2 ± 4.1	31.7 ± 5.8	29.7 ± 4.9 *
Systolic blood pressure (mm Hg)	141 ± 18	139 ± 18	142 ± 19	139 ± 17
Pulse pressure (mm Hg)	64 ± 17	66 ± 17	65 ± 17	64 ± 17
Glycemia (mg/dL)	118 ± 38	123 ± 42	116 ± 35	122 ± 42
Total cholesterol (mg/dL)	189 ± 42	166 ± 32 **	186 ± 43	181 ± 40
Non-HDL cholesterol (mg/dL)	141 ± 41	117 ± 30 **	138 ± 40	132 ± 39
Triglycerides (mg/dL)	119 ± 45	117 ± 54	121 ± 51	116 ± 44
Statin user (*n* (%))	46 (48)	26 (74) *	31 (52)	41 (59)
Serum creatinine (mg/dL)	1.5 ± 0.4	1.8 ± 0.5 ***	1.5 ± 0.4	1.6 ± 0.4
Serum uric acid (mg/dL)	6.3 ± 1.5	6.6 ± 1.5	6.3 ± 1.6	6.4 ± 1.4
GFR (mL/min/1.73 m^2^)	48 ± 15	42 ± 16 *	52 ± 15	42 ± 15 ***
24-h urinary albumin (mg/day)	23 (6.8–29)	24 (11–29)	23 (6–29)	23 (9–29)
24-h urinary protein (mg/day)	120 (68–150)	120 (86–173)	117 (78–155)	120 (69–151)
Longitudinal kidney diameter (mm)	10.1 ± 1.2	10.1 ± 1.2	10.1 ± 1.4	10.0 ± 1.2
Kidney cortical thickness (mm)	1.4 ± 0.2	1.3 ± 0.2	1.4 ± 0.2	1.4 ± 0.3
Intrarenal RI	0.74 ± 0.03	0.83 ± 0.03 ***	0.75 ± 0.04	0.78 ± 0.05 **

* *p* < 0.050; ** *p* < 0.010; *** *p* < 0.001; Cr., creatinine; RI, resistive index; Tia, transitory ischemic attack; GFR, glomerular filtration rate by creatinine clearance.

**Table 3 jcm-11-02995-t003:** Different models analyzed by mixed-effect linear regression of variables associated with serum creatinine levels change during follow-up.

	Model 1	Model 2	Model 3
Estimate	Coefficient (95%CI)	Coefficient (95%CI)	Coefficient (95%CI)
Intercept	1.47 (1.36, 1.58)	1.26 (1.13, 1.39)	1.25 (1.13, 1.36)
Time (years)	0.02 (0.00, 0.04) *	0.02 (0.00, 0.04) *	0.02 (0.00, 0.05) *
Intrarenal RI ≥ 0.80 (yes/no)	0.28 (0.07, 0.49) *	0.21 (0.01, 0.40) *	0.12 (−0.06, 0.30)
Interaction: intrarenal RI × time	0.06 (0.02, 0.10) **	0.06 (0.02, 0.11) **	0.06 (0.02, 0.11) **
Age (years)—scaled	-	0.08 (0.00, 0.16)	0.01 (−0.07, 0.08)
Male sex (yes/no)	-	0.40 (0.24, 0.56) ***	0.47 (0.33, 0.62) ***
GFR—scaled	-	-	−0.23 (−0.31, −0.16) ***

* *p* < 0.050; ** *p* < 0.010, *** *p* < 0.001. CI, confidence interval; RI, resistive index; GFR, glomerular filtration rate by creatinine clearance. Intercept, regression line intercept; Interaction, interaction term in the regression analysis; Scaled, variable was mean centered and divided by standard deviation.

**Table 4 jcm-11-02995-t004:** Univariate and best logistic regression model for renal disease progression assessed by at least 0.5 mg/dL increment of creatinine levels during 5-year follow-up.

	Univariate		Best Model	
Variable	OR (95% CI)	*p*	OR (95% CI)	*p*
Age (each year)	1.05 (1.00–1.12)	0.060	-	-
Male sex (yes/no)	1.31 (0.59–2.97)	0.515	-	-
Past or active smoker (yes/no)	1.05 (0.38–2.68)	0.921	-	-
Diagnosis of (yes/no): Hypertension Diabetes Coronary artery disease Heart failure Atrial fibrillation Tia/stroke Peripheral artery disease	2.16 (0.55–14.4) 1.14 (0.47–2.67) 4.89 (1.74–14.2) 3.44 (1.29–9.17) 3.71 (1.32–10.5) 0.29 (0.04–1.07) 0.84 (0.12–3.69)	0.330 0.761 0.003 0.013 0.013 0.107 0.832	- 2.48 (0.74–8.63) 14.7 (3.51–77.0) 3.10 (0.88–11.4) 7.01 (1.76–30.5) 0.21 (0.03–1.04) -	- 0.143 <0.001 0.080 0.007 0.088 -
Body mass index (each kg/m^2^)	0.94 (0.86–1.01)	0.111	0.49 (0.17–1.30)	0.170
SBP (each 10 mm Hg)	0.96 (0.76–1.20)	0.743	-	-
Pulse pressure (each 10 mm Hg)	1.10 (0.87–1.39)	0.420	-	-
Glycemia (each 10 mg/dL)	1.02 (0.92–1.12)	0.699	-	-
Total cholesterol (each 10 mg/dL)	0.99 (0.90–1.10)	0.918	1.21 (1.04–1.43)	0.017
Non-HDL cholesterol (each 10 mg/dL)	0.98 (0.89–1.09)	0.735	-	-
Triglycerides (each 10 mg/dL)	1.06 (0.97–1.15)	0.188	-	-
Statin user (yes/no)	1.36 (0.61–3.09)	0.452	-	-
Serum creatinine (each mg/dL)	1.41 (0.57–3.48)	0.455	-	-
Serum uric acid (each mg/dL)	1.00 (0.76–1.31)	0.995	-	-
GFR (each 10 mL/min/1.73 m^2^)	0.83 (0.63–1.07)	0.161	-	-
24-h urinary albumin (each log mg/day)	2.08 (1.18–4.04)	0.018	-	-
24-h urinary protein (each log mg/day)	2.89 (1.44–6.30)	0.006	3.95 (1.56–11.9)	0.008
Longitudinal kidney diameter (each mm)	1.41 (1.01–2.01)	0.050	-	-
Kidney cortical thickness (each mm)	0.81 (0.14–4.31)	0.801	-	-
Intrarenal RI (each 0.1)	2.83 (1.26–6.67)	0.013	4.13 (1.45–12.9)	0.010

HR, hazards ratio; CI, confidence interval; SBP, systolic blood pressure; GFR, glomerular filtration rate by creatinine clearance; RI, resistive index; Tia, transitory ischemic attack.

**Table 5 jcm-11-02995-t005:** Univariate and best Cox proportional hazards regression model for all-cause mortality.

	Univariate		Best Model	
Variable	HR (95% CI)	*p*	HR (95% CI)	*p*
Age (each year)	1.12 (1.07–1.17)	<0.001	1.11 (1.06–1.15)	<0.001
Male sex (yes/no)	1.43 (0.89–2.32)	0.143	-	-
Past or active smoker (yes/no)	1.07 (0.60–1.90)	0.810	-	-
Diagnosis of (yes/no): Hypertension Diabetes Coronary artery disease Heart failure Atrial fibrillation Tia/stroke Peripheral artery disease	0.87 (0.43–1.75) 0.76 (0.45–1.29) 1.97 (1.07–3.61) 2.15 (1.21–3.83) 1.75 (0.94–3.28) 1.20 (0.64–2.24) 1.60 (0.73–3.50)	0.694 0.315 0.030 0.009 0.079 0.566 0.241	- 0.58 (0.33–1.02) 1.78 (0.94–3.35) 1.93 (1.04–3.57) - - -	- 0.060 0.075 0.036 - - -
Body mass index (each kg/m^2^)	0.94 (0.90–0.99)	0.019	0.94 (0.89–0.99)	0.013
SBP (each 10 mm Hg)	0.94 (0.82–1.08)	0.379	-	-
Pulse pressure (each 10 mm Hg)	0.99 (0.86–1.14)	0.913	-	-
Glycemia (each 10 mg/dL)	1.05 (0.99–1.12)	0.113	1.07 (1.01–1.14)	0.032
Total cholesterol (each 10 mg/dL)	0.96 (0.91–1.02)	0.184	-	-
Non-HDL cholesterol (each 10 mg/dL)	0.96 (0.90–1.02)	0.145	-	-
Triglycerides (each 10 mg/dL)	1.00 (0.95–1.05)	0.899	-	-
Statin user (yes/no)	1.38 (0.85–2.22)	0.192	-	-
Serum creatinine (each mg/dL)	1.71 (1.01–2.87)	0.044	-	-
Serum uric acid (each mg/dL)	1.06 (0.90–1.24)	0.506	-	-
GFR (each 10 mL/min/1.73 m^2^)	0.74 (0.62–0.87)	<0.001	-	-
24-h urinary albumin (each log mg/day)	1.26 (0.93–1.71)	0.138	-	-
24-h urinary protein (each log mg/day)	1.06 (0.75–1.49)	0.754	-	-
Longitudinal kidney diameter (each mm)	0.94 (0.77–1.16)	0.579	-	-
Kidney cortical thickness (each mm)	0.71 (0.25–2.02)	0.525	-	-
Intrarenal RI (each 0.1)	2.37 (1.44–3.90)	<0.001	1.80 (1.05–3.09)	0.034

HR, hazards ratio; CI, confidence interval; SBP, systolic blood pressure; GFR, glomerular filtration rate by creatinine clearance; RI, resistive index; Tia, transitory ischemic attack.

## Data Availability

The datasets generated and analyzed during the current study are available from the corresponding author on reasonable request and after authorization by the Administrative Department of the University of Udine.

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
