# Peer review of "Elevated Intrarenal Resistive Index Predicted Faster Renal Function Decline and Long-Term Mortality in Non-Proteinuric Chronic Kidney Disease"

_jcm, 2022, doi:10.3390/jcm11112995_

Round 1
Reviewer 1 Report
This study focuses on the relationship between intrarenal resistive index (RI) and renal outcomes in non-proteinuric chronic kidney disease (CKD). The study has clinically practical significance. However, the following revision is recommended.
- Study design: Please provide a flow diagram of participants.
- Results: CKD is associated with a 1.5- to 3.5-fold increased risk for cardiovascular disease, previous studies have found that the decline of eGFR in CKD patients is related to the increased arterial wall inflammation, the thickening of the carotid artery wall, and plaque formation [J Am Soc Nephrol, PMID: 27799487], and reduced renal function has been shown to be independently associated with lower cerebral blood flow [J Am Soc Nephrol, PMID: 26251352]. If possible, the authors may provide the Carotid Doppler data of the participants, and analyze it with renal function, intrarenal resistive index, and other indicators.
- Discussion: Studies on the relationship between biopsy-confirmed kidney disease and RI have yielded conflicting results. Some authors found that only arteriosclerosis was independently associated with RI among all histological parameters, while others found glomerulosclerosis and tubulointerstitial disease to be the strongest predictors of RI [Renal Failure, PMID: 31599199]. Please further discuss the formation and influencing factors of RI.
- Others: there are format problems in the manuscript, such as inconsistent fonts, etc. [Line 181-182; Table 2].
- The present version of the manuscript is not well readable and may be edited by a professional English Editing Service.
Reviewer 2 Report
The authors aimed to evaluate the role of intrarenal RI in predicting the progression of renal disease and long-term mortality in patients with non-proteinuric CKD of unknown origin. The idea to evaluate the role of intrarenal RI for renal outcome is not fully new. But the study is well performed and presented. To increase readability and quality of the presentation I suggest to add study flowchart, ROC analysis and present progression of CKD as renal survival K-M curves (dialysis-free outcome, or doubling of creatinine, or increase of creatinine by 0.5 mg/dl). Please add "years" in Figure 3. It is misleading now.
Round 2
Reviewer 2 Report
Thank you for your corrections. The manuscript was improved. It was shown that intrarenal RI was an independent predictor of renal function decline and mortality, but it had poor discrimination performance. It is worth to note that in the described cohort of patients risk of death was higher than risk of kidney function deterioration. I suggest to correct the sentence in the abstract "Patients with intrarenal RI ≥0.80 had a faster annual increase of serum creatinine levels (+0.06 mg/dl, 95% CI 0.02-0.10, p<0.010) compared to those with RI<0.80 (+0.06 mg/dl, 95% CI 0.02-0.10, p<0.010)." It is unclear. In table 2 stating use was presented as mean and SD instead of proportion.
